# Using Zebrafish Animal Model to Study the Genetic Underpinning and Mechanism of Arrhythmogenic Cardiomyopathy

**DOI:** 10.3390/ijms24044106

**Published:** 2023-02-18

**Authors:** Yujuan Niu, Yuanchao Sun, Yuting Liu, Ke Du, Xiaolei Xu, Yonghe Ding

**Affiliations:** 1The Biomedical Sciences Institute of Qingdao University (Qingdao Branch of SJTU Bio-X Institutes), Qingdao University, Qingdao 266021, China; 2School of Public Health, Qingdao University, Qingdao 266021, China; 3Department of Biochemistry and Molecular Biology, Mayo Clinic, Rochester, MN 55902, USA

**Keywords:** arrhythmogenic cardiomyopathy, zebrafish, genetic, disease model

## Abstract

Arrhythmogenic cardiomyopathy (ACM) is largely an autosomal dominant genetic disorder manifesting fibrofatty infiltration and ventricular arrhythmia with predominantly right ventricular involvement. ACM is one of the major conditions associated with an increased risk of sudden cardiac death, most notably in young individuals and athletes. ACM has strong genetic determinants, and genetic variants in more than 25 genes have been identified to be associated with ACM, accounting for approximately 60% of ACM cases. Genetic studies of ACM in vertebrate animal models such as zebrafish (*Danio rerio*), which are highly amenable to large-scale genetic and drug screenings, offer unique opportunities to identify and functionally assess new genetic variants associated with ACM and to dissect the underlying molecular and cellular mechanisms at the whole-organism level. Here, we summarize key genes implicated in ACM. We discuss the use of zebrafish models, categorized according to gene manipulation approaches, such as gene knockdown, gene knock-out, transgenic overexpression, and CRISPR/Cas9-mediated knock-in, to study the genetic underpinning and mechanism of ACM. Information gained from genetic and pharmacogenomic studies in such animal models can not only increase our understanding of the pathophysiology of disease progression, but also guide disease diagnosis, prognosis, and the development of innovative therapeutic strategies.

## 1. Introduction

Arrhythmogenic cardiomyopathy (ACM) is an updated term for the heart muscle disorder originally known as arrhythmogenic right ventricular dysplasia/cardiomyopathy (ARVD/C), manifesting as malignant ventricular arrhythmias and fibrofatty infiltration in the myocardium of both ventricles, with right ventricular involvement being more common. Depending on the patient cohorts studied, the prevalence of ACM is estimated to be 1 case per 2000 to 5000 individuals in the general population [1]. While recognized as a rare form of cardiac disease, ACM with sustained ventricular arrhythmia represents a major risk condition, causing a significant proportion of sudden cardiac deaths, especially in young individuals and athletes [2,3].

In the 2000s, through pedigree linkage analysis, homozygous truncating mutations in the junction plakoglobin (*JUP*) and desmoplakin (*DSP*) genes were identified in parallel to cause ACM that overlaps with cardio-cutaneous syndromes, representing the first two genetic studies to recognize ACM as an inheritable type of cardiomyopathy [4,5]. Since then, over the past almost two decades, mostly through the combined approaches of familial genetic linkage and candidate gene sequencing, numerous pathogenic variants in more than 25 genes have been identified to cause ACM, accounting for up to 60% of ACM cases [6,7,8,9]. These findings underscore the strong genetic determinant of ACM. However, the genotype–phenotype relationship in ACM patients remains complex; even for ACM cases harboring the same identified gene mutations, disease onset and disease severity can manifest with highly variable expressivity, likely due to the contributions of other comorbidities combined with different genetic background and environmental factors [10,11]. In addition, for patients with known genetic causes, the molecular mechanisms underlying their ACM disease progression remain poorly understood. Currently, clinical management of ACM patients mainly focuses on the general control of heart failure and the prevention of arrhythmic sudden cardiac death (SCD) [12]. No specific genotype-tailored therapeutic approaches for ACM are currently available.

Classic human genetic studies, such as familial linkage analysis, are limited by exhaustible pedigree resources and difficulties in obtaining appropriate tissue samples, while next-generation genome-wide association studies (GWASs) lack the power to directly define the genotype–phenotype relationship. In addition, human genetic studies are also affected by other intrinsic limitations, including different lifestyles and environmental stimuli such as exercise, inflammation, viral infection, and drug usage, which hampered advances in our understanding of the ACM disease pathophysiology and the development of effective therapies.

As animal models can be housed under standardized and controlled conditions, the potential influences of many confounding variables such as environmental factors and genetic background can be circumvented to help define the causal roles of gene mutations in ACM. For variants with defined causal roles, genetic animal models allow for ready investigation of the molecular and cellular mechanisms underlying ACM disease progression at the whole organism level. Indeed, studies using the development of in vivo animal models combined with in vitro cell culture systems have led to the discovery of several major signaling pathways, such as the Wnt/β-catenin, Hippo/Yes associated protein (YAP), and transforming growth factor-β (TGF-β) signaling pathways, which are involved in the pathogenesis of ACM [8,9,13]. A deeper understanding of disease pathophysiology and mechanisms of action would help to identify new therapeutic targets and pave the way for the development of innovative disease therapies and cures. 

Traditionally, owing to their well-conserved physiology to humans, mammals such as rodents have been the gold standard for animal models when studying disease pathophysiology and the underlying disease mechanisms of action. Due to the relatively long and costly experimental process and, thus, low throughput, rodent models are limited when functionally validating a vast number of gene variants and discovering new genetic disease associations on a large scale. Another intrinsic limitation when using mice to model cardiac arrhythmia diseases such as ACM is that their average heart rate of approximately 500 beats per minute (bpm), which is much higher than the average human heart rate of approximately 60–90 bpm. In contrast, zebrafish (*Danio rerio*) have an average heart rate of ~100 bpm and share very similar cardiac electrophysiology features with humans. In this regard, zebrafish are justified as an attractive animal model for studying cardiac arrhythmia diseases such as ACM [14,15,16].

Zebrafish are small tropical vertebrates that share 87% genetic similarity with humans. Its embryonic transparency, high fecundity, and conserved cardiac electrophysiology in comparison with humans make it a popular animal model for studying development of the cardiovascular system and heart function and for cardiotoxicity drug screening [17]. Recently, with technique advances such as the optimized electrocardiogram (ECG) for analysis of cardiac electrophysiology, voltage and calcium mapping for measurement of action potential and calcium transients, and noninvasive high-frequency echocardiography (HFE) for cardiac function analysis, zebrafish have emerged as a prolific animal model to study the genetic basis and underlying mechanisms of ACM [18,19,20]. To date, several pharmacological and genetic screening studies using zebrafish as the animal model for ACM have been performed, aiming to identify and develop novel drug targets [21,22]. These genetic studies combined with high-throughput chemical screening in the zebrafish model have just begun to identify some promising targets that can potentially be developed to exert therapeutic effects on ACM [23]. 

In this review, we briefly summarize the genetic causes of ACM. We highlight several key mechanistic insights into ACM disease pathophysiology resulting from genetic studies in animal models, including zebrafish. We focus on the use of zebrafish models to study the genetic basis of ACM, categorized according to the approaches employed: targeted gene knockdown (KD)/knock-out (KO), transgenic overexpression, and knock-in (KI) gene editing. Finally, we envisage using the zebrafish animal model amenable to large-scale genetic and chemical screening, in conjunction with highly efficient CRISPR/Cas9-based precision gene editing tools, to streamline functional validation and pharmacogenomic studies that can potentially inform innovative therapeutic strategies for ACM. 

## 2. Genetic Determinants of ACM

Since the first identification of junction plakoglobin (*JUP*) and desmoplakin (*DSP*) gene mutations linked to ACM in the 2000s, to date, hundreds of variants in more than 25 genes have been identified to be associated with ACM, which can explain up to 60% of inherited ACM cases, underscoring the strong genetic determinants of ACM [6,7,8,9,24]. Based on their subcellular localization and function, these 25 ACM associated genes can be classified into distinct categories, including desmosome, adherens junction, sarcomere and Z-disc, integrin binding, sarcoplasmic reticulum, cytoskeletal filamin, nuclear and nuclear envelope, sodium channel, and cytokine genes (Table 1 and Figure 1).

Desmosomes are multiple-protein complexes responsible for mediating cell adhesion and signal transduction [59,60]. Thus, it is plausible to believe that the structural integrity of desmosomes is critical to maintain normal cardiac function and contraction rhythm. A large body of human genetic studies have found that approximately half of patients with ACM have one or multiple mutations identified in desmosomal genes, including desmoplakin (*DSP*) [5], plakophilin-2 (*PKP2*) [25], desmoglein-2 *(DSG2*) [28], and desmocollin-2 (*DSC2*) [31], encoding the main components of the desmosome. *PKP2* gene mutations have been identified in up to 45% of ACM patients, representing the most commonly affected gene [25,61]. Aside from desmosomes, there are other connections between myocardial cells; investigating these connections has led to the identification of mutations in adherens junction genes, including cadherin 2 (*CDH2*) [37,38] and catenin-α3 (*CTNNA3*) [35], in patients with ACM. 

In addition to the classic desmosomal and adherens junction genes, mutations in other nonjunctional genes have also been identified sporadically. These ACM-associated genes encode a variety of proteins localized to the sarcomeric z-discs, sarcoplasmic reticulum, nucleus and nuclear envelope, and cytoplasm and are involved in a wide array of biological processes such as ion transport, calcium regulation, maintaining nuclear and sarcomere structural integrity, cytoskeletal filament binding, and cytokine secretion (Table 1 and Figure 1).

Although classically viewed as a monogenic disorder, the genetic culprits of ACM are very diverse. Increasing evidence implicates the contributions of compound heterozygosity and digenic or even oligogenic mutations and the possible coexistence of modifier genes [26,62,63]. According to the current research on the full array of genetic variants in ACM cases, truncation mutations are the most common type. Nonsense mutations, frameshift mutations, splice site mutations, and missense variants were also frequently identified. In most cases, ACM is typically inherited as autosomal dominant with highly variable phenotypic expression and incomplete penetrance. Nevertheless, autosomal recessive mutations, most notably in cardio-cutaneous forms such as Naxos disease and Carvajal syndrome, have also been documented [9,33,64].

## 3. Molecular Pathways Involved in the Pathogenesis of ACM

The key features of ACM pathogenesis manifest as adipogenesis and myocardium replacement, fibrosis, and ventricular arrhythmia. As mutations in desmosome account for the majority of inherited ACM cases, ACM is traditionally considered a disease of desmosomes. In the heart, desmosomes are part of the unique structure connecting myocardial cells called intercalated disks. When desmosome components-encoding genes are mutated, the adhesion between cells is decreased, and the intercalated disks are destabilized, causing abnormal biomechanical remodeling and ion channel dysregulation, which subsequently contributes to the pathogenesis of ACM. To date, based on combinatory studies in the in vitro cell culture systems such as human-induced pluripotent stem cell (hiPSC) and in vivo animal models, including the zebrafish, our understanding of molecular pathways underlying the pathogenic mechanisms of ACM has greatly advanced. Several key signal pathways, such as Wnt/β-catenin pathway, Hippo/YAP pathway, and TGF-β signaling pathways, have been implicated (Figure 2) [8,9,13].

Wnt ligand binding to the Frizzled receptor and its co-receptor LRP5 and LRP6, known as canonical Wnt/β-catenin signaling, is a highly conserved and complex protein action network that regulates key cellular functions from embryonic heart development to adult heart physiological processes [65]. A large body of evidence has supported that interference with Wnt/β-catenin signaling plays a central role in the adipogenesis and fibrosis of desmosomal ACM [66,67,68]. In the TMEM43 p.S358L mutant (TMEM43-S358L) overexpressing mouse model of ACM, sustained activation of glycogen synthase kinase-3β (GSK3β), a direct downstream regulator of Wnt/β-catenin signaling, appeared to be the primary pathogenic trigger causing cardiomyocyte death and replacement by fibrofatty tissues [69]. Chemical treatment with the GSK3β inhibitor SB216763 improved cardiac function and prolonged lifespan in TMEM43-S358L mutant overexpressing mice. Plakoglobin, also known as γ-catenin, has a similar structure and function to β-catenin and can compete with Wnt ligand binding. In cultured atrial myocytes deficient in the desmosome gene *DSP*, plakoglobin protein was released from factor binding in the cytosol and translocated to the nucleus, resulting in elevated expression of adipogenic and fibrogenesis-related genes, thus contributing significantly to the fibrofatty infiltration phenotypes of ACM [67].

Interestingly, in the zebrafish model of *Dsp*-deficiency induced ACM, three pathways, including Wnt/β-catenin, TGF-β, and Hippo/YAP, were significantly affected, among which Wnt/β-catenin signaling was noted to be altered the most dramatically. Supported by evidence from both genetic and pharmacological studies, Wnt/β-catenin was further proposed to be the final common pathway underlying the pathogenic mechanisms of desmosomal gene mutation-based ACM [68].

In addition to Wnt/β-catenin signaling, dysregulation of the Hippo/YAP and TGF-β signaling pathways is also implicated in the pathogenesis of ACM. The Hippo signaling pathway is evolutionarily conserved across species and plays a key role in controlling cardiomyocyte cell proliferation and heart size [70]. The Hippo pathway exerts its role through a series of kinase phosphorylation reactions and by interacting with the downstream effectors of Yes-associated protein (YAP) and transcriptional coactivator with PDZ-binding motif (TAZ) [71,72]. Mounting evidence has demonstrated a link between enhanced adipogenesis and pathogenic activation of the Hippo pathway, concomitant with suppression of Wnt/β-catenin signaling in human hearts from patients with ACM and in animal models of ACM resulting from desmosomal gene deficiency [68,73]. In fact, while the components of the Hippo and Wnt/β-catenin pathways are known to be interactive and intertwined in controlling cardiomyocyte proliferation [74], the crosstalk between YAP and catenin-β1 causes Hippo activation and suppression of catenin-β1 nuclear translocation, which is thought to directly correlate with lipogenesis in the Pkp2-depleted HL-1 cell system [73]. 

The TGF-β signaling pathway represents another molecular event implicated in pathogenesis, most notably in the induction of myocardial fibrosis in ACM [75,76]. TGF-β signaling is known to suppress the expression of matrix metalloproteinases (MMPs) while increasing the expression of tissue inhibitors of matrix metalloproteinase (TIMP) to stimulate fibroblasts to differentiate into myofibroblasts [77]. In cardiomyocytes of the *JUP* KO mouse model of ACM, the cardiac fibrosis and spontaneous ventricular arrhythmia phenotypes were associated with significantly elevated TGFβ-1 and phosphor-SMAD2, but not Wnt/β-catenin signaling in the early stages of ACM phenotype manifestation [78]. In addition, in the neonatal rat ventricular myocyte (NRVM) *Pkp2*-KD model of ACM, TGFβ-1-dependent expression of fibrotic genes was also significantly increased [79]. Thus, this reported experimental evidence supports a pathogenic role of TGF-β signaling in the progressive fibrosis of desmosomal gene mutation-induced ACM. Intriguingly, variants in the secreted protein TGFβ3 encoding gene were also identified in two family cases of ACM, providing additional evidence to support the role of TGF-β paracrine signaling in the ACM pathogenesis. While the exact cellular connection whereby malfunctions of desmosomes trigger alternation in paracrine signaling warrants further investigation, a recent study in mice model of ACM caused by the epicardial cell-specific *Dsp* gene deletion suggests contributions of epicardial-derived cells to the pathogenesis of ACM [80].

## 4. Zebrafish Models of ACM Genes

Because of the intrinsic limitations of human genetic studies, animal models are needed to gain dynamic insights into disease pathophysiology, to explore underlying molecular and cellular mechanisms, and to test potentially targeted therapeutic strategies. Because of the amenability of zebrafish to large-scale genetic and chemical screening, boosted by advanced cardiac phenotypic tools such as electrocardiogram (ECG) [20], voltage and calcium mapping and high-frequency echocardiography (HFE) [18,19], zebrafish have emerged as a prolific animal model for studying the genetic basis and pharmacogenomics of ACM. To model hereditary ACM in animal models such as zebrafish, there are typically 4 different gene manipulation approaches, including gene KD and KO of the ACM-associated genes, transgenic overexpression of mutated ACM variants, and KI of pathogenic variants in ACM genes that are practically implemented.

### 4.1. Transient KD Models of ACM Genes

Gene-targeted inactivation strategies involve global KD or KO of gene function in animals, representing one of the most convenient and effective approaches to model human diseases caused by genetic mutations, resulting in loss-of-function. Thus, KD/KO-mediated gene inactivation methods are most often used to create animal models to study loss-of-function mutations that mostly exhibit autosomal recessive genetic patterns. 

In the early 1990s, morpholino antisense technology was developed as a very effective in vivo tool in zebrafish for transient KD of gene expression during early embryogenesis [17]. In the early stages, the majority of zebrafish models for ACM genes employed the morpholino-induced KD technique. In general, morpholino-induced KD of desmosomal genes in zebrafish often resulted in significant defects in cardiogenesis [31,68,81,82]. For example, zebrafish injected with the *pkp2* morpholino, also known as morphants, developed cardiac edema, heart looping defects, reduced heart rate, and altered desmosome structure [82]. Morphants of other desmosomal genes such as *DSC2*, *JUP,* or *DSP* in zebrafish also exhibited heart edema, bradycardia, and reduced contractility [68,81,82]. Notably, morphants of the corresponding zebrafish orthologs of the human *DSP* gene (*dspa* and/or *dspb*) displayed reduced and disorganized desmosomal junctions, leading to mild developmental delay phenotypes. At the molecular level, specific alterations in the Wnt/β-catenin, TGF-β, and Hippo/YAP signaling pathways were identified through an in vivo cell signaling screen. Among the three altered pathways identified, Wnt/β-catenin signaling was most dramatically suppressed, and this suppression was rescued by either *dspa* mRNA expression or by treatment with the Wnt agonist SB216763. This study thus supported the value of zebrafish as a useful animal model for detecting early signaling pathways underlying the pathogenesis of DSP-associated ACM diseases that can potentially be the basis for novel pharmacological therapies [68].

In addition to the desmosomal genes, KD of the zebrafish orthologous genes corresponding to other nondesmosome ACM genes, such as *ILK, LMNA, SCN5A, ACTN2, FLNC,* and *DES,* using the morpholino technique, has also been studied. Most of these morphants usually exhibited defects during early heart developmental stages and, in some cases, with a high incidence of arrhythmias accompanied by varying degrees of muscular defects (Table 2) [83,84,85,86,87,88,89].

While zebrafish morphant models are convenient to generate and may provide certain disease pathogenic insights into the absence of specific ACM gene functions, morpholino-related techniques are restricted to transient KD/inactivation of gene expression during the early embryonic stage. Since ACM pathogenesis seems to be more complicated than expected and many ACM patients develop slowly progressing phenotypes that do not manifest symptoms until the adult stages, it is necessary to have inactivate gene function beyond the embryonic stages to model ACM disease progression and allow for more comprehensive investigation of disease pathophysiology and underlying mechanisms. 

### 4.2. Stable KO Models of ACM Genes

New techniques, such as zinc finger nucleases (ZFNs), transcription activator-like effector nucleases (TALENs) and, more recently, CRISPR/Cas9-based genomic editing, have revolutionized the zebrafish gene editing toolbox, which enables the generation of stable gene KO mutants, a routinely available technique accessible to even a small laboratory [17]. To date, several ACM-associated genes, including *TTN* and *TMEM43,* have been modeled in zebrafish using the gene KO technique (Table 2). 

The *TTN* (Titin) gene encodes the largest sarcomeric protein in the human body. While *TTN* truncating variants (*TTNtvs*) are considered the most common cause of familial dilated cardiomyopathy (DCM), they can also account for up to 10% of ACM cases [9,39,95]. To model *TTNtv*-associated cardiomyopathy diseases in zebrafish, a series of *TTNtv* mutants were created using TALEN-based gene editing technology. While homozygous *TTNtv* mutants developed severe cardiac defects, ventricular chamber enlargement, reduced heart rate, and premature death [91,92], heterozygous *TTNtv* mutants were viable adults but exhibited contractile dysfunction upon hemodynamic stress concomitant with enlarged atrial chamber size phenotypes. This study implies that hemodynamic stress might be an important contributing factor to *TTNtv*-related DCM and atrial arrhythmogenesis [91].

The *TMEM43* gene encodes transmembrane protein 43, which is mostly localized to the inner nuclear membrane and is important for maintaining nuclear envelope structural integrity [96]. Several rare *TMEM43* variants have been sporadically identified in ACM patients [53,54,55,58,97]. To investigate the molecular mechanism underlying *TMEM43* mutation-associated ACM, a truncation mutant was generated by using the CRISPR/Cas9-based gene editing technique to shift the reading frame of the zebrafish *tmem43* gene [94]. Interestingly, significantly enlarged ventricular chamber size was observed in the loss-of-function *tmem43* mutants in the adult stage, while no obvious cardiac defects or contractile dysfunction were detected during early embryonic stages. 

Global homozygous KO of genes function might lead to embryonic lethality in some cases, such as in those with the *TTN* gene KO mutant, which precludes disease modeling in adults. More emphasis is needed on heterozygous or haploinsufficient mutants, which can be more representative of genetic mutations found in human patients. To circumvent early embryonic lethality, an alternative method is to generate conditional and/or tissue-specific KO models and/or disrupt gene function in zebrafish models, which allows us to characterize the mutant phenotype and functionally assess the ACM gene and the underlying pathophysiological and molecular mechanisms in a temporally and spatially restricted manner.

### 4.3. Transgenic Overexpression Models of ACM Variants

Targeted KD/KO of ACM genes can mostly mimic large deletion or truncating mutations that exert loss-of-function or haploinsufficiency effects when mutated in human patients. However, many missense mutations found in human ACM patients often cause gain-of-function or dominant-negative/toxic effects. In these scenarios, transgenic overexpression of patient-derived mutant alleles in animal models provides a more suitable platform to mimic and explore disease pathophysiology and the underlying molecular and cellular mechanisms for variants exhibiting autosomal dominant inheritance patterns. Advances in transgenic approaches, such as Tol2 transposase mediated transgenesis, have made it highly efficient and reliable to overexpress gene variants identified in ACM patients in zebrafish models. To date, several human ACM gene variants in genes, including *SCN5A*, *JUP*, *ILK*, and *TMEM43*, have been modeled in zebrafish through transgenic overexpression approaches (Table 3). 

The *SCN5A* gene encodes the voltage-gated cardiac sodium channel protein type 5 subunit alpha, also known as Nav1.5, which plays a pivotal role in impulse propagation through the heart. *SCN5A* mutations are associated with a variety of cardiac phenotypes such as Brugada syndrome, sick sinus syndrome, cardiac conduction disease, dilated cardiomyopathy, and ACM [98,99]. The prevalence of pathogenic *SCN5A* variants in ACM cases was estimated to be approximately 2% [57]. In 2013, a cardiac-specific transgenic line harboring the human *SCN5A* D1275N variant in a zebrafish model was generated and characterized [100]. This *SCN5A* D1275N transgenic zebrafish line exhibited mostly cardiac conduction system abnormalities, such as increased incidence of sinus pause, atrioventricular block, bradycardia, and prolonged PR interval and QRS duration. Together with morpholino KD on two of the corresponding zebrafish *SCN5A* ortholog genes displaying severe heart dysmorphogenesis phenotype and death during embryonic stages, these zebrafish models of the *SCN5A* gene recapitulated certain aspects of arrhythmia phenotypes and cardiac structural abnormalities in human ACM patients.

**Table 3 ijms-24-04106-t003:** Transgenic overexpression and knock-in models of ACM gene variants in zebrafish.

Human ACM Gene	Zebrafish Ortholog	Model Type	Variant	Zebrafish Phenotype	Reference
*SCN5A*	*scn5a*	Tg/OE	human *SCN5A* D1275N	Conduction defects, increased beat-to-beat variations, bi-ventricular cardiomyopathy, reduced survival	[20,100]
	*scn5b*				
*JUP*	*jupa*	Tg/OE	Human *JUP* 2057del2	Heart enlargement, reduced action potential, reduced I_Na_ and I_K1_ current density, decreases survival	[22]
	*jupb*				
*ILK*	*ilk*	Tg/OE	Human *ILK* H77Y, H33N, P70L	Cardiac dysfunction, reduced action potential, epicardial fat deposit, premature death	[45]
*TMEM43*	*tmem43*	Tg/OE	Human *TMEM43* S358L, P111L	Enlarged hearts with cardiomyocyte hyperplasia, cardiac dysfunction	[94]
*PLN*	*plna*	KI/CRISPR	Zebrafish *plna* R14del	Heart size enlargement, sub-epicardial inflammation and fibrosis, lipid accumulation, increased beat-to-beat variation in cardiac output	[21]

KI, knock-in; Tg/OE, transgenic/over expression.

*JUP,* encoding a junction plakoglobin protein that is a major component of desmosomes, is one of the first two genes identified to associate with ACM. It was originally identified in patients from a Greek island who presented with woolly hair and palmoplantar keratoderma, known as Naxos disease [4,33]. Later, it was linked to ACM and has been extensively studied in different model systems mostly mice and rats [7,8]. Because a homozygous two-nucleotide deletion mutation in the *JUP* gene (*JUP*-2507del2) was frequently identified in patients with ACM, a cardiac-specific transgenic line overexpressing the human *JUP*-2507del2 mutation in zebrafish to model *JUP* gene-based ACM was generated [22]. This transgenic ACM model line manifested enlarged ventricular chamber size with marked thinning of atrial and ventricular walls, changes in action potential with reduced I_Na_ and I_K1_ current density, and decreased survival, recapitulating several key features of human ACM. A more detailed mechanistic study was performed to indicate that altered trafficking of Nav1.5, Kir2.1, and plakoglobin proteins to the intercalated disc might be the pathologic mechanism underlying *JUP*-2507del2-induced ACM. This zebrafish ACM model was further optimized by introducing a luciferase reporter, and a screenable zebrafish model of ACM for high-throughput drug screening was created. Following f 4200 small molecules screened, a single chemical, SB216763, that could rescue the arrhythmia phenotypes of ACM, and prevent heart failure and reduce mortality in this fish model, was identified. These therapeutic effects of SB216763 on ACM phenotypes were further confirmed in in vitro cultured neonatal rat ventricular myocytes and iPSCs derived from ACM probands with mutations in the plakophinlin-2 gene. SB216763 is a highly selective glycogen synthase kinase-3 (GSK-3) inhibitor [101], and its therapeutic effects on ACM were proposed to be achieved at least partially by activating the canonical Wnt signaling pathway by inhibiting GSK-3β. This study provides a remarkable example of the use of zebrafish models to illuminate new ACM disease mechanisms and discover mechanism-based anti-arrhythmic drugs that can suppress or ameliorate ACM disease phenotypes.

Integrin-linked kinase *(ILK)* encodes a serine/threonine kinase that exhibits both enzymatic and scaffolding roles in integrin-mediated signal transduction. In 2019, two novel missense *ILK* variants (p.H33N and p.H77Y) in two unrelated ACM families were identified through exome sequencing [45]. Functional validation studies were performed through generating transgenic overexpression zebrafish models of *ILK* H33N, H77Y, and the P70L variant that was previously identified in a patient with DCM, driven by the cardiomyocyte-specific cardiac myosin light chain 2 (*cmlc2)* promoter. The results showed that cardiac-specific overexpression of the *ILK* H77Y and P70L variants led to cardiac dysfunction and heart failure, resulting in premature death in zebrafish. Intriguingly, moderate to severe epicardial fat accumulation was further detected in the *ILK* H77Y and P70L variants overexpressing fish hearts, recapitulating the key feature of the ACM phenotype in human patients. Zebrafish harboring cardiac-specifical overexpression of the *ILK* H33N variant survived to adulthood but exhibited impaired action potential defects. This study thus demonstrated the causality of these three variants in the inherited cardiomyopathy diseases, providing an excellent example of use zebrafish as an in vivo animal to functionally validate and characterize ACM variants identified from human genetic studies. 

For the *TMEM3* gene, in addition to the aforementioned KO study, transgenic overexpression study on two different ACM-associated *TMEM43* variants (P111L and S358L) was also reported [94]. Transgenic overexpression of *TMEM43* gene, either wild-type or mutated (P111L), resulted in cardiac defects such as cardiomyocyte hypertrophy and ventricular chamber enlargement during the early developmental stages in zebrafish. Mechanistically, it was proposed that overexpression of *TMEM43* variants induced cardiac defects were mediated by hyperactivation of the mammalian target of rapamycin (mTOR) pathway and ribosome biogenesis. This combinatorial study, using both gene KO and transgenic overexpression approaches, indicated that both gain-of-function and loss-of-function of the *TMEM43* gene led to cardiac morphology and function defects in zebrafish [94]. Thus, these findings might imply a potentially unique pathophysiological feature of *TMEM43*-associated ACM in human patients. 

By greatly overexpressing mutant variants in animal models in vivo, the transgenic overexpression approach allows us to investigate pathogenic and molecular insights into protein deposition mechanisms. However, since the overexpressed mutated protein is not subject to endogenous gene regulation, it might be expressed at different levels compared to those in the physiological state and exhibit different expression patterns compared to the endogenous protein; therefore, it is not strictly physiologically relevant, which might lead to pleiotropic effects that are different from those in human patients carrying the same mutation. Moreover, zebrafish genetically modified through a transgenic overexpression approach cannot model a human disease caused by biallelic mutations. 

### 4.4. KI Models of ACM Alleles

KI models involve generation of a one-for-one substitution or specific insertion/deletion of gene variants into the endogenous gene locus in animals, referred to as a targeted DNA sequence substitution/deletion/insertion. Because most human-inheritable ACM diseases are caused by alleles that carry single nucleotide changes, resulting in altered gene function, KI thus represents a much more faithful system to better recapitulate missense mutations in human ACM patients. It can model inheritable ACM diseases caused by both loss-of-function and gain-of-function mutations.

While the efficiency and reliability of KI technique in zebrafish models can be further improved, many studies have demonstrated the feasibility of using the CRISPR/Cas9-based precision gene editing approach to generate KI variants for a variety of genes including those implicated in cardiovascular diseases [102,103]. In 2018, a KI protocol by combining the CRISPR/Cas9 with a short template oligonucleotide was optimized and used to generate 4 different KI zebrafish lines carrying gain-of-function mutations in 4 human ortholog genes to model human genetic cardiovascular disorders [104]. One of the four KI lines, phospholamban (*PLN*) p.Arg14del (R14del), was subsequently followed-up and reported as the first patient-specific zebrafish model for ACM (Table 3) [21]. The human *PLN* gene encodes a 52-amino acid transmembrane sarcoplasmic reticulum protein that regulates the calcium (Ca^2+^) pump in cardiomyocyte [105]. *PLN* mutations, including the R14del deletion, have been identified in up to 15% of Dutch patients with DCM and/or ACM [106], and in cardiomyopathy patients in other countries as well [107,108]. This zebrafish model of *PLN* R14del ACM exhibited age-related cardiac remodeling concurrent with subepicardial inflammation, fibrosis, and lipid accumulation, resembling key features of human ACM patients. More detailed phenotypic analysis revealed contractile variation and action potential duration alternation ahead of overt cardiac structural remodeling. At the cellular and molecular levels, a reduction in Ca^2+^ transient amplitudes, but an increase in the diastolic Ca^2+^ level in adult cardiomyocytes, was appreciated before the detection of functional impairment at the cardiac organ level. Based on these detailed pathophysiological findings mostly related to Ca^2+^ dysregulation, the drug istaroxime targeting sarcoplasmic reticulum (SR) Ca^2+^ sequestration was subsequently tested. The results showed that istaroxime treatment could restore Ca^2+^ regulation, ameliorate the impaired action potential duration, and ultimately improve cardiac dysfunction. Thus, this study represents a remarkable example of utilizing a KI model to explore the underlying pathophysiological mechanisms of ACM that could be leveraged to develop novel-targeted therapeutic drugs. 

## 5. Limitations

While most of the clinical features of ACM, such as ventricular arrhythmia, myocardial chamber dilation, and sudden cardiac death, can be recapitulated in zebrafish models, one major limitation in using zebrafish to model ACM is that replacement of the myocardium with fibrofatty tissue, a hallmark of ACM, is uncommon in zebrafish models, and this is also the case for mouse models [7]. To date, the lipid accumulation phenotype has been detected only on two occasions: one is the *ILK* H77Y and P70L variants overexpressing zebrafish model of ACM [79], and the other is the *PLN* R14del KI zebrafish [21].The insufficiency to recapitulate fibrofatty infiltration phenotype in mouse and zebrafish models is likely due to intrinsic species differences between zebrafish and humans in which the zebrafish heart does not have comparable epicardial fat to that seen in the human heart. To address this bottleneck, the development of ACM models in large animals, such as dogs or cats, might be worth consideration, as most of the key features of ACM, including ventricular arrhythmia, ventricular structural remodeling, and fibrofatty infiltration in the myocardium, seem to be recapitulated well in boxer dogs and domestic cats [109,110,111,112].

Other limitations when using zebrafish to model human ACM include the following: (1) A zebrafish heart consists of only a two-chamber myocardium structure, including one ventricle and one atrium, which is different from the four-chamber structure in humans. In addition, the zebrafish heart ventricle is a sponge-like structure without a clearly-defined ventricular wall, as humans have. Thus, it is challenging to model the right, left, or biventricular structural remodeling phenotypes in human ACM; (2) Zebrafish have experienced an additional whole-genome duplication event, resulting in two co-orthologs for ~15% of corresponding human genes [113]. Therefore, the potential gene compensation and/or function redundancy effects need to be taken into account when modeling the human ACM; (3) Zebrafish maintain cardiomyocyte regeneration capacity throughout the lifecycle, which is different from humans, in which CM regeneration capacity mostly occurs in the fetal stages and is considered lost after birth. Thus, the contribution of potential cardiomyocyte proliferation to ACM disease progression needs to be considered.

## 6. Conclusions and Perspectives

Studies in zebrafish models have helped to elucidate the genetic underpinning and advance our understanding of ACM pathophysiology. New drugs that have the potential to cure ACM have just begun to be identified [21,22]. Despite certain shortcomings of zebrafish models, to overcome some of the intrinsic limitations of human genetic studies, we anticipate combined utilization of the prolific animal models of zebrafish in the following perspectives to streamline the genetic and pharmacogenomic study of ACM. 

### 6.1. Assessment of the Functional Consequences of Sequence Variants

While hundreds of variants in more than 25 genes have been identified to be associated with ACM in the past two decades, the number of genomic variants and genes associated with ACM continues to expand at an unprecedented rate in the so-called next-generation sequencing era. For the vast number of genetic variants identified mostly through human genetic studies, however, systematically assessing their functional significance and implications remains a major challenge. With the advent of powerful CRISPR/Cas9 gene-editing approaches, it is now technically highly efficient to induce precise base conversion and introduce specific patient alleles in the corresponding zebrafish orthologs on a reasonable scale [114]. Animal models such as zebrafish, which are highly amenable to genetic manipulations and precision gene editing, have a high-throughput capacity and thus provide instrumental tools to functionally validate and annotate the potential impacts of a large number of variants of unknown significance (VUSs) on ACM disease progression in vivo. For patient alleles with confirmed disease causative roles, genetic animal models allow for investigation of the molecular and cellular mechanisms underlying ACM disease progression at the whole organism level. 

### 6.2. Study of Oligogenic or Multiple Gene Interactions

Similar to other hereditary heart diseases, it has been increasingly recognized that some ACM patients may carry multiple compound pathogenic variants, raising the possibility of a digenic or oligogenic basis for ACM [26,62,63,115]. The coexistence of combined interactions among multiple causative and/or modifier genes can at least partially explain the highly variable disease penetrance and phenotypic expressivity, highlighting the complex genetic mechanisms underlying the pathogenesis of ACM. With advances in CRISPR/Cas9-based gene editing tools, it is now very convenient to create conditional and tissue-specific biallelic mutations for multiple genes in the F0 generation in zebrafish models [116,117,118]. To experimentally prove the model of a possible digenic or oligogenic basis for ACM, zebrafish models that are highly amenable to genetic manipulations can thus offer many advantageous opportunities to systematically study genetic interactions among multiple gene variants in a short time frame. The resultant findings can thus help to determine the complex genotype–phenotype relationships more accurately and move toward a better understanding of ACM disease pathophysiology and the development of new therapeutic targets.

### 6.3. Forward Genetic Screening to Identify New ACM Elusive Genes

While a strong genetic determinant of ACM has been demonstrated, there are no pathogenic variants identified in up to 40% of familial ACM patients. Identification of a complete list of all ACM disease-associated genes and variants is critical to obtain a higher genetic testing yield, which can not only help to improve diagnostics, but can also provide better prognostic value to inform tailored management. For example, mutations in the DSP and PLN genes are often associated with worse arrhythmia and a higher risk of sudden cardiac death [26,63], and patients harboring the TMEM43 S358L variant are predicted to have high ACM disease penetrance and severe arrhythmic risk [119]. However, due to limited and exhaustible patient pedigree resources, identification of pathogenic variants in genes of elusive patients remains a challenge. To overcome these limitations in human genetics, unbiased forward genetic screening in zebrafish enabled by highly efficient transposon-mediated mutagenesis appears to be an attractive strategy for systematic identification of novel ACM causative genes, as exemplified by the identification of the *SORBS2* gene as a new ACM associated gene with human relevance, which was initially identified through forward mutagenesis screening in adult zebrafish [44].

### 6.4. Chemical Screening to Develop New Therapeutic Drugs

In addition to its advantages in forward mutagenesis screening approaches, zebrafish are also amenable to high-throughput chemical screening, offering attractive opportunities for systematic ACM susceptibility gene discovery and the identification of novel therapeutic small molecular modulators. As exemplified by a study by Asimaki and coauthors who screened 4200 small molecules using a zebrafish model of ACM generated through transgenic overexpression of the *PKP2* 2057del2 gene variant in zebrafish larvae, and SB216763 was identified as a potent drug that could reverse the ACM phenotypes in this animal model [22]. Another example is the identification of istaroxime as a potential therapeutic drug for ACM that can ameliorate Ca^2+^ abnormalities, an early pathophysiological event underlying ACM disease progression in models with the *PLN* p.Arg14del KI [21]. These encouraging achievements through combinatory studies of mechanistic investigation, coupled with high-throughput chemical screening enabled by zebrafish models, have just begun to identify some promising drug targets that can be potentially developed to exert therapeutic effects on ACM.

In summary, we envisage utilizing complementary approaches embracing animal models, such as zebrafish, and human genetics to study the genetic basis of ACM and to investigate the underlying molecular and cellular mechanisms. By incorporating animal models into high-throughput drug screening pipelines, we expect that the resultant findings would provide a better understanding of ACM disease pathogenesis and move toward the identification of new therapeutic drug targets. Knowledge gained from these complementary studies could subsequently be further leveraged to guide the development of better diagnosis and prognosis, and to inform innovative therapeutic strategies.

## Figures and Tables

**Figure 1 ijms-24-04106-f001:**
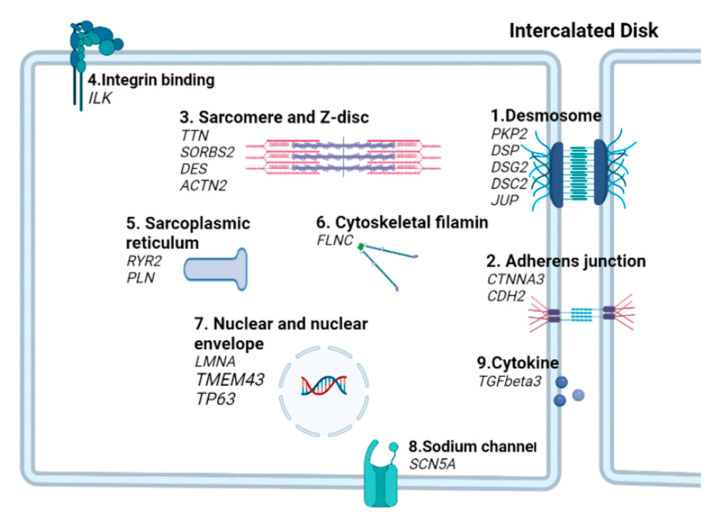
Schematic illustration of subcellular components of ACM genes.

**Figure 2 ijms-24-04106-f002:**
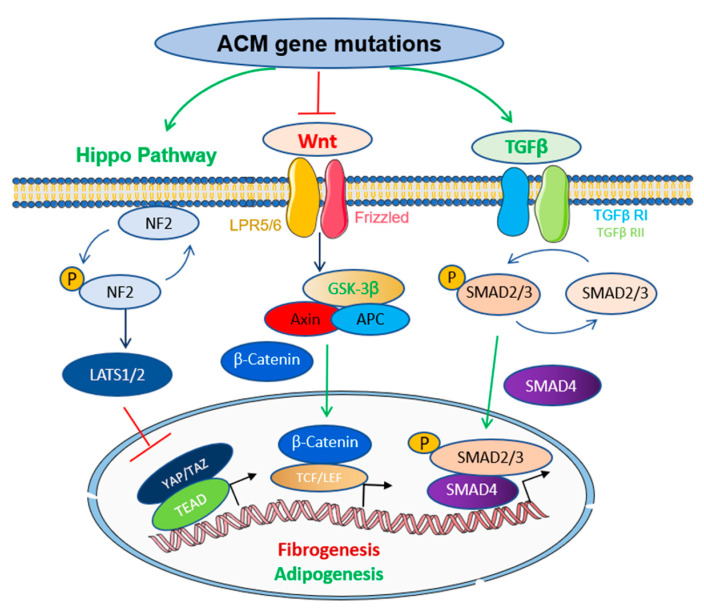
Major signaling pathways involved in the pathogenesis of inherited ACM.

**Table 1 ijms-24-04106-t001:** Summary of key genes implicated in ACM.

Gene (Protein)	Gene Ontology (GO) Annotation	Estimated Frequency in ACM	Mutation Type	Mode of Inheritance	Phenotypic Features	References
Desmosome						
*PKP2* (Plakophilin-2)	Cell–cell adhesion; intermediate filament bundle assembly, heart development	20–45%	Splice-site, nonsense, ins/del, large del, missense	Mostly AD	RV dominant ACM, DCM	[25,26,27]
*DSG2* (Desmoglein-2)	Component of intercellular desmosome junctions, calcium ion binding	3–20%	Splice-site, nonsense, ins/del, missense	AD and AR	Biventricular ACM, overlap with DCM	[28,29]
*DSP* (Desmoplakin)	Cell–cell adhesion, intermediate filament cytoskeleton organization	3–20%	Splice-site, nonsense, ins/del, large del, missense	AD and AR	Overlap with Carvajal syndrome, frequent LV involvement, overlap with DCM	[5,30]
*DSC2* (Desmocollin-2)	Component of intercellular desmosome junctions, cadherin binding, cell adhesion, anchoring junction	1–15%	Splice-site, nonsense, ins/del, missense	AD and AR	BiVCM, LV and RV involvement	[31,32]
*JUP* (Plakoglobin)	Common junctional plaque protein, cadherin binding, cell adhesion, anchoring junction	0–1%	Splice-site, nonsense, ins/del, missense	AR	Overlap with Naxos disease, RV dominant	[33,34]
Adherens junction						
*CTNNA3* (Catenin alpha 3)	Involved in formation of stretch-resistant cell–cell adhesion complexes, beta-catenin binding, cell migration	0–1%	Missense, del	AD	RV involvement with ACM	[35,36]
*CDH2* (Cadherin-2)	Cell–cell adhesion via plasma–membrane adhesion molecules, catenin complex, calcium ion binding	0–2%	Missense	AD	RV and LV involvement with ACM	[37,38]
Sarcomere and Z-disc						
*TTN* (Titin)	A key component in the assembly and functioning of muscles, protein serine/threonine kinase activity, calcium ion binding	0–10%	Missense, truncation	AD	Overlap with DCM	[39,40]
*DES* *(Desmin)*	Located in Z disc, intermediate filament organization	0–3%	Missense, del	AD	LV dominant ACM, DCM, and skeletal myopathies	[41,42]
*ACTN2* (Actin alpha 2)	Actin filament binding, actin cytoskeleton organization; calcium ion binding	A single case	Missense	AD	Left ventricular dominant	[43]
*SORBS2* (Sorbin and SH3 domain containing 2)	Actin filament organization, focal adhesion	2 cases	Missense	AD	Overlap with DCM, associated with atrial septal defects	[44]
Integrin binding						
*ILK* (Integrin-linked kinase)	Involvement of integrin-mediated signal transduction	2 cases	Missense	AD	Overlap with DCM	[45]
Sarcoplasmic reticulum						
*RYR2* (Ryanodine receptor 2)	Calcium channel activity	0–10%	Missense	AD	Right ventricular involvement	[46]
*PLN* (Phospholamban)	Negative regulation of ATPase-coupled calcium transmembrane transporter activity	0–3%	deletion	AD	Overlap with DCM	[47,48]
Cytoskeletal filamin						
*FLNC* (Filamin C)	Actin cytoskeleton organization, actin filament binding	0–3%	Missense, nonsense, splicing or frameshifts	AD	LV dominant ACM	[49,50]
Nuclear and nuclear envelope						
*LMNA* (Lamin A/C)	Intermediate filament proteins, scaffolding components of the nuclear envelope	3–4%	Missense, nonsense	AD	Overlap with DCM, muscular dystrophy	[51,52]
*TMEM43* (Transmembrane protein 43)	Inner Nuclear membrane organization, protein self-association	Sporadic	Missense	AD	Overlap with Emery-Dreifuss muscular dystrophy, male more severely affected than female	[53,54,55]
*TP63* (Tumor Protein-63)	DNA-binding transcription factor activity, RNA polymerase II-specific	Single case	Missense	AD	Ectodermic dysplasia, adult sudden death	[56]
Sodium channel						
*SCN5A* (Nav1.5)	sodium voltage-gated channel, electrical signal regulation and contraction coordination	0–2%	Missense	AD	Overlap with Brugada syndrome, RV involvement with ACM, long QT syndrome	[57]
Cytokine						
*TGFB3* (transforming growth factor-β3)	Type III transforming growth factor beta receptor binding, positive regulation of pathway-restricted SMAD protein phosphorylation	2 cases	Missense	AD	RV, LV dominant ACM, DCM and skeletal myopathies	[58]

ACM, arrhythmogenic cardiomyopathy; AD, autosomal dominant; AR, autosome recessive; ARVC, arrhythmogenic right ventricular cardiomyopathy; BiVCM, biventricular arrhythmia; DCM, dilated cardiomyopathy; RV, right ventricle; LV, left ventricle; HCM, hypertrophic cardiomyopathy.

**Table 2 ijms-24-04106-t002:** Knockdown/knock-out animal models of ACM genes in zebrafish.

ACM Gene	Zebrafish Ortholog	Model Type	Mutation	Zebrafish Phenotype	Reference
*DSC2*	*dsc2*	KD/MO	Translation inhibiting and exon skipping	Cardiac edema, contractile dysfunction, myocardium remodeling	[31]
*ILK*	*ilk*	KO/ENU	L308P	Heart contractility defects	[88]
	*ilk*	KD/MO	Translation inhibiting and splice-blocking	Cardiac ejection capacity decreased	
*JUP*	*plakoglobin*	KD/MO	Translation inhibiting	Small hearts, cardiac edema, and valvular dysfunction	[81]
*LMNA*	*lamin A/C*	KD/MO	Translation inhibiting	Bradycardia, cardiomyopathy	[85]
	*lamin A*	KD/MO	Splice-blocking	Embryonic senescence, lipodystrophy, Muscle and Cartilage Abnormalities	[90]
*SCN5A*	*scn5Laa*	KD/MO	Translation inhibiting and splice-blocking	Chamber formation defects	[87]
	*scn5Lab*	KD/MO			
	*scn5Laa; scn5Lab*	KD/MO		Cardiac defects more severe than either gene alone	
*PKP2*	*plakophilin 2*	KD/MO	Translation inhibiting	Cardiac edema, heart looping defects, reduced heart rate, altered desmosome structure	[82]
*ACTN2*	*actn2*	KD/MO	Exon skipping and translation inhibiting	Cardiac and skeletal muscle defects	[84]
*FLNC*	*flnca*	KD/MO	Translation inhibiting	Slow muscle fiber failure, myosin containing globules formed at the muscular septum	[86]
	*flncb*	KD/MO	Exon skipping	Muscle fibers break along the length of the fiber, rather than detach from one end	
	*flnca;flncb*	KD/MO	Translation inhibiting and exon skipping	Fast muscle defects, Myosin aggregation, fiber failure	
	*flncb*	KO/ENU	Frameshift, premature stop	Slow muscle fibers failure, myosin form aggregation	
*DES*	*desma;* *desmb*	KD/MO	Splice-blocking	Cardiac edema, muscle dis-integrity	[83]
*DSP*	*dspa*	KD/MO	Translation inhibiting	Reduced and disorganized desmosomal junctions	[68]
	*dspa;* *dspb*	KD/MO	Translation inhibiting/Exon skipping		
*TTN*	*ttn.1^−/−^*	KO/TALEN	Frameshift, truncation	Myofibrils disarrangement, premature death	[91,92]
	*ttn.2^−/−^*	KO/TALEN	Frameshift, truncation	Ventricular enlargement, reduced heart rate, cardiac dysfunction, premature death	
	*ttn.2^tv/+^*	KO/TALEN	Frameshift, truncation	Cardiac dysfunction, atrial chamber enlargement, decreased sarcomere content	
	*ttn.1^/−^;ttn.2^−/−^*	KO/TALEN	Frameshift, truncation	Fish paralyzed, severely disrupted myofibril, premature death,	
*TP63*	*tp63^−/−^*	KO/CRISPR	Frameshift, premature stop	Ectoderm-derived structures defects, embryonic lethal, no obvious cardiac phenotypes detected	[93]
*TMEM43*	*Tmem43^−/−^*	KO/CRISPR	Frameshift, truncation	Late-onset ventricular enlargement	[94]

KD, knockdown; MO, morpholino; KO, knock-out; ENU, N-ethyl-N-nitrosourea; TALEN, transcription activator-like-effector nuclease; CRISPR, clustered regularly interspaced short palindromic repeats.

## Data Availability

Not applicable.

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
