# Peer review of "Using Zebrafish Animal Model to Study the Genetic Underpinning and Mechanism of Arrhythmogenic Cardiomyopathy"

_ijms, 2023, doi:10.3390/ijms24044106_

Round 1

Reviewer 1 Report

In this review, Niu et al summarized key genes implicated in arrhythmogenic cardiomyopathy (ACM) and in detail described the use of zebrafish models to study the underlying mechanisms of ACM. Overall, the contents of this review are of particular interest to the general audient as well as experts in this field, the writing is clear and logical. There are several minor suggestions to improve its quality for publication.

1. It seems to me the writing style is split and not consistent. The first part was precise and in passive voice, while the second part (from section 4.2) was in active voice (“the authors found…”) and very detailed. This resulted in several very long paragraphs which were not convenient for the readers. I will suggest reducing descriptions of experiment details but focusing on the conclusion and significance.

2. The order and categories of genes in the main text, figures and tables should be consistent. For examples, the numeric order and categories in Figure 1 was different from Table 1 (ILK and FLNC were in one category in the table but separated in the figure). The authors intended to describe zebrafish models of ACM genes based on the genetic manipulation in section 4 and Table 2/3, however, they organized the text as one paragraph for one gene with mixed approaches (KO with transgenesis for tmem43, transgenesis with KD for scn5, etc). Splitting the paragraphs may be better.

3. There are many typos or grammar errors, the authors should carefully revise the manuscript or seek help from a professional editing service. For examples:

-Line 22 “approach” should be in plural form.

-Line 91 “in compared with” had an error.

-Table 1 “cytoskeletal filamin” had a typo.

-Line 160-161, “In the heart, desmosomes are part of the unique structure connecting myocardial cells.” better add “called intercalated disks” at the end.

-Line 171, “Wnt ligand binding to the β1-catenin” is not scientifically correct.

-Table 2, “scn5Lab; scn5Lab” should be aa;ab.

4. The reference citation had some errors. In Table 2, the reference for TTN should be 64,65, not 65,66. The reference for tp63 should be 66. In the text, 68,69 first appeared earlier than 64,65. And 67 first appeared after 75.

Author Response

In this review, Niu et al summarized key genes implicated in arrhythmogenic cardiomyopathy (ACM) and in detail described the use of zebrafish models to study the underlying mechanisms of ACM. Overall, the contents of this review are of particular interest to the general audient as well as experts in this field, the writing is clear and logical. There are several minor suggestions to improve its quality for publication.

  1. 1. It seems to me the writing style is split and not consistent. The first part was precise and in passive voice, while the second part (from section 4.2) was in active voice (“the authors found…”) and very detailed. This resulted in several very long paragraphs which were not convenient for the readers. I will suggest reducing descriptions of experiment details but focusing on the conclusion and significance.

Response: Thank you very much for pointing out this logic flaw for our manuscript. We edited the second part from section 4.2 by switching to mostly passive voice for those ACM genes studied in zebrafish models, making it more consistent with the first part of this manuscript. While we continue to focus more on the conclusion and significance, we hope you will agree that we still keep those experimental details, as we believe some of those details are helpful to enrich the major topic of this manuscript on using zebrafish models to study genetic aspects and mechanisms of ACM.

  1. The order and categories of genes in the main text, figures and tables should be consistent. For examples, the numeric order and categories in Figure 1 was different from Table 1 (ILK and FLNC were in one category in the table but separated in the figure). The authors intended to describe zebrafish models of ACM genes based on the genetic manipulation in section 4 and Table 2/3, however, they organized the text as one paragraph for one gene with mixed approaches (KO with transgenesis for tmem43, transgenesis with KD for scn5, etc). Splitting the paragraphs may be better.

Response: We really appreciate these great suggestions. We edited the manuscript accordingly and made it more consistent between the main text and Figures and Tables. We spit the paragraphs for each of the genes or genetic manipulation approaches as suggested.

  1. There are many typos or grammar errors, the authors should carefully revise the manuscript or seek help from a professional editing service.

Response: Thank you very much for pointing out our typos and wording issues. We corrected all the pointed typos and reworded the corresponding sentences based on your suggestions.  We also double checked the whole manuscript and corrected other typos and grammar issues accordingly.

Before the initial submission, this manuscript was edited already for proper English language, grammar, punctuation, spelling, and overall style by one or more of the qualified native English-speaking editors at the Springer Nature Author Services (SNAS) (We’d happy to provide a copy of the editing certificate if requested). Because of this, we hope you are OK we are not sending it out for additional editing service.

For examples:

-Line 22 “approach” should be in plural form.

Response: We corrected it to “approaches”.

-Line 91 “in compared with” had an error.

Response: We corrected them to “in comparison with”.

-Table 1 “cytoskeletal filamin” had a typo.

Response: We corrected this typo accordingly.

-Line 160-161, “In the heart, desmosomes are part of the unique structure connecting myocardial cells.” better add “called intercalated disks” at the end.

Response: We edited this sentence as suggested.

-Line 171, “Wnt ligand binding to the β1-catenin” is not scientifically correct.

Response: We edited this sentence to “Wnt ligand binding to the Frizzled receptor and its co-receptor LRP5 and LRP6, known as canonical Wnt/β-catenin signaling, …”.

-Table 2, “scn5Lab; scn5Lab” should be aa;ab.

Response: We corrected this typo accordingly.

  1. The reference citation had some errors. In Table 2, the reference for TTN should be 64,65, not 65,66. The reference for tp63 should be 66. In the text, 68,69 first appeared earlier than 64,65. And 67 first appeared after 75.

Response: Thank you very much for pointing out our reference citation errors. We corrected them accordingly. We have also tried our best to make sure all other references are cited appropriately.

Reviewer 2 Report

Summary: This review article summarizes current knowledge regarding genes that underlie arrhythmogenic cardiomyopathy (ACM), how zebrafish has contributed to that knowledge, and the strengths and limitations the zebrafish model offers for future work in deciphering functional consequences in mutations of ACM genes. It is well written and informative.

Comments:

·       Line 158-159 points out that mutations in desmosome and gap junction genes account for the majority of inherited ACM cases, and later sentences describe a cellular connection between desmosome and gap junction gene functions. However, gap junction mutations are not mentioned elsewhere in the manuscript as a genetic cause for ACM. Why not?

·       It is quite interesting that mutations in desmosomal mutations trigger downstream dysregulation of paracrine signaling pathways (Figure 2), and that mutations in genes of those pathways also cause ACM.  Is anything known about the cellular connection whereby malfunction of desmosomes would trigger alterations in paracrine signaling?

·       The identification of small molecule SB216763 as an inhibitor of JUP-based ACM in a sensitized zebrafish screen was an innovative advance coming directly via this model system. After being published in 2014, was SB216763 ever investigated further in whole-animal mammalian or human trials as a potentially marketable therapeutic?

·       Table 1 describes some of the variable phenotypes of ACM, linked to various genes. Can anything be said about the gene expression patterns (if spatially and/or temporally constrained), and the tendency for the phenotype to be left or right ventricle predominant? This may not be possible from the zebrafish data since zebrafish have only the single ventricle.

·       Line 183: It is somewhat ambiguous to say gamma-catenin is “released”, which (to me, anyway) suggests a release to the outside of the cell. It would be helpful to reword this to be clear that gamma-catenin is released from factors binding it in the cytosol with the result that it translocates to the nucleus.

·       Line 437: the phrase “restore calcium abnormalities” is a bit unclear, because it sounds like the abnormalities are brought back again, when the authors mean the opposite; “restore calcium handling”, or “restore calcium regulation” may be better.

Minor comments

·       Line 18: should be “are highly amenable to large-scale genetic…”

·       Line 78: “it is justified zebrafish” à zebrafish are justified as an attractive animal model…

·       Line 91: “in compared with” à in comparison with

·       Note there is a typo in Cytoskeleton Filamin in Table 1

·       Line 134: “investigating these connects” à investigating these connections

·       Line 196” “revolutionarily conserved”? I imagine this should be “evolutionarily”

·       Line 200: “demonstrated a link”

·       Line 212: “increase” à increasing

·       Line 246: “also known as morphants…”

·       Line 250: dspb needs italics but no underline

·       Line 300: typo in CRISPR

·       Line 303: redundant use of “adult”

·       Line 324: “temporally and spatially manner” à temporally and spatially restricted manner

·       Line 356: patients from a Greek island

·       Line 439: should be “utilizing a KI model”

Author Response

Summary: This review article summarizes current knowledge regarding genes that underlie arrhythmogenic cardiomyopathy (ACM), how zebrafish has contributed to that knowledge, and the strengths and limitations the zebrafish model offers for future work in deciphering functional consequences in mutations of ACM genes. It is well written and informative.

Comments:

 Line 158-159 points out that mutations in desmosome and gap junction genes account for the majority of inherited ACM cases, and later sentences describe a cellular connection between desmosome and gap junction gene functions. However, gap junction mutations are not mentioned elsewhere in the manuscript as a genetic cause for ACM. Why not?

Response: We are sorry for causing this confusion. We meant “mutations in desmosome genes account for the majority of inherited ACM cases”. We thus deleted the “gap junction genes” from this sentence. 

It is quite interesting that mutations in desmosomal mutations trigger downstream dysregulation of paracrine signaling pathways (Figure 2), and that mutations in genes of those pathways also cause ACM.  Is anything known about the cellular connection whereby malfunction of desmosomes would trigger alterations in paracrine signaling?

Response: Indeed, mutations in desmosomal mutations trigger downstream dysregulation of paracrine signaling pathways, notably dysregulation of Wnt and TGF-β superfamily proteins (Figure 2), and mutations in genes of these pathways such as TGFβ3 are associated with rare ACM cases (Table 1).

To the best of our knowledge, the cellular connection whereby malfunctions of desmosomes trigger alternation in paracrine signaling is not well documented in existing literature reports so far. Nevertheless, a recent study through conditional Dsp gene deletion in the epicardial cells coupled with single-cell RNA sequencing analysis in mice model provides some insights into the contributions of the epicardial-derived cells to the pathogenesis of desmosomal gene mutation-induced ACM (Yuan et al., Circulation, 2021) 1. This study concludes that “Epicardial cells express desmosome proteins and deletion of Dsp, encoding desmoplakin, in the epicardial cells leads to myocardial fibro–adipogenesis, apoptosis, cardiac dysfunction, and premature death. Dsp haploinsufficient epicardial cells give rise to endothelial cells (ECs), fibroblasts, epithelial cells that express and secret paracrine factors, such as TGFβ1, which mediate epithelial-mesenchymal transition (EMT) and contribute to the pathogenesis of cardiac phenotype in ACM”.

Reference:

  1. Yuan P, Cheedipudi SM, Rouhi L, Fan S, Simon L, Zhao Z, Hong K, Gurha P and Marian AJ. Single-Cell RNA Sequencing Uncovers Paracrine Functions of the Epicardial-Derived Cells in Arrhythmogenic Cardiomyopathy. Circulation. 2021;143:2169-2187.

To appreciate this insightful comment from you, we edited the manuscript by adding the text: “Intriguingly, variants in the secreted protein TGFβ3 encoding gene were identified in two cases of ACM, providing additional evidence to support the role of TGF-β paracrine signaling in the ACM pathogenesis. While the exact cellular connection whereby malfunctions of desmosomes trigger alternation in paracrine signaling warrants further investigation, a recent study in mice model of ACM induced by the epicardial cell-specific Dsp gene deletion suggests contributions of the epicardial-derived cells to the pathogenesis of ACM” (lines 226-232).

The identification of small molecule SB216763 as an inhibitor of JUP-based ACM in a sensitized zebrafish screen was an innovative advance coming directly via this model system. After being published in 2014, was SB216763 ever investigated further in whole-animal mammalian or human trials as a potentially marketable therapeutic?

Response: Yes, since its identification as a potential therapeutic drug for JUP-based ACM model in zebrafish, the protective effect of SB216763 on ACM phenotypes was further validated in mouse models in vivo, in human induced pluripotent stem cells (hiPSC) derived cardiomyocytes in vitro, and in human cardiac slices (Asimaki et al., Cir Arrhythm Electrophysiol,2016; Chelko et al., JCI insight, 2016; Li et al., JACC:  JACC Basic Transl Sci. 2022; Padron-Barthe, Circulation, 2019;) 2-5. However, this SB216763 small molecule has not reached the clinical trial or market yet.

References

  1. Asimaki A, Protonotarios A, James CA, Chelko SP, Tichnell C, Murray B, Tsatsopoulou A, Anastasakis A, te Riele A, Kleber AG, Judge DP, Calkins H and Saffitz JE. Characterizing the Molecular Pathology of Arrhythmogenic Cardiomyopathy in Patient Buccal Mucosa Cells. Circ Arrhythm Electrophysiol. 2016;9:e003688.
  2. Chelko SP, Asimaki A, Andersen P, Bedja D, Amat-Alarcon N, DeMazumder D, Jasti R, MacRae CA, Leber R, Kleber AG, Saffitz JE and Judge DP. Central role for GSK3beta in the pathogenesis of arrhythmogenic cardiomyopathy. JCI Insight. 2016;1.
  3. Li G, Brumback BD, Huang L, Zhang DM, Yin T, Lipovsky CE, Hicks SC, Jimenez J, Boyle PM and Rentschler SL. Acute Glycogen Synthase Kinase-3 Inhibition Modulates Human Cardiac Conduction. JACC Basic Transl Sci. 2022;7:1001-1017.
  4. Padron-Barthe L, Villalba-Orero M, Gomez-Salinero JM, Dominguez F, Roman M, Larrasa-Alonso J, Ortiz-Sanchez P, Martinez F, Lopez-Olaneta M, Bonzon-Kulichenko E, Vazquez J, Marti-Gomez C, Santiago DJ, Prados B, Giovinazzo G, Gomez-Gaviro MV, Priori S, Garcia-Pavia P and Lara-Pezzi E. Severe Cardiac Dysfunction and Death Caused by Arrhythmogenic Right Ventricular Cardiomyopathy Type 5 Are Improved by Inhibition of Glycogen Synthase Kinase-3beta. Circulation. 2019;140:1188-1204.

Table 1 describes some of the variable phenotypes of ACM, linked to various genes. Can anything be said about the gene expression patterns (if spatially and/or temporally constrained), and the tendency for the phenotype to be left or right ventricle predominant? This may not be possible from the zebrafish data since zebrafish have only the single ventricle.

Response: It is certain that some of the ACM implicated genes exhibit spatially and temporally constrained expression patterns at both the whole-organ and single cell levels. It is also clinically recognized that the genotype of ACM mutation carriers has significant impact on phenotypic expression and disease severity. For example, mutations in the DSP and PLN genes are often associated with worse arrhythmia and a higher risk of sudden cardiac death, and patients harboring the TMEM43 S358L variant are predicted to have high ACM disease penetrance and severe arrhythmic risk. However, to the best of our knowledge, a direct link and/or correlation between the gene expression patterns and the tendency of the phenotype to be left or right ventricle predominant is not literarily established yet.

We agree with you that since zebrafish have only one single ventricle, it is almost impossible to model the right, left or biventricular structure remodeling phenotypes of human ACM in zebrafish. We listed this limitation in the text (lines 492-496).

 Line 183: It is somewhat ambiguous to say gamma-catenin is “released”, which (to me, anyway) suggests a release to the outside of the cell. It would be helpful to reword this to be clear that gamma-catenin is released from factors binding it in the cytosol with the result that it translocates to the nucleus.

Response: Thanks for pointing out this inappropriate wording usage. We made changes according to your suggestion.

Line 437: the phrase “restore calcium abnormalities” is a bit unclear, because it sounds like the abnormalities are brought back again, when the authors mean the opposite; “restore calcium handling”, or “restore calcium regulation” may be better.

Response: Thanks for pointing out this inappropriate wording usage. We changed it to “restore calcium regulation”.

Minor comments

  • Line 18: should be “are highly amenable to large-scale genetic…”
  • Line 78: “it is justified zebrafish” à zebrafish are justified as an attractive animal model…
  • Line 91: “in compared with” à in comparison with
  • Note there is a typo in Cytoskeleton Filaminin Table 1
  • Line 134: “investigating these connects” à investigating these connections
  • Line 196” “revolutionarily conserved”? I imagine this should be “evolutionarily”
  • Line 200: “demonstrated a link”
  • Line 212: “increase” à increasing
  • Line 246: “also known as morphants…”
  • Line 250: dspb needs italics but no underline
  • Line 300: typo in CRISPR
  • Line 303: redundant use of “adult”
  • Line 324: “temporally and spatially manner” à temporally and spatially restricted manner
  • Line 356: patients from a Greek island
  • Line 439: should be “utilizing a KI model”

Response: Thank you very much for pointing out the above typos and grammar issues, we corrected all of them accordingly.
